# 13-Type HPV DNA Test versus 5-Type HPV mRNA Test in Triage of Women Aged 25–33 Years with Minor Cytological Abnormalities–6 Years of Follow-Up

**DOI:** 10.3390/ijerph20054119

**Published:** 2023-02-25

**Authors:** Amir Rad, Sveinung Wergeland Sørbye, Tormod Brenn, Sweta Tiwari, Maja-Lisa Løchen, Finn Egil Skjeldestad

**Affiliations:** 1Department of Community Medicine & Department of Clinical Medicine, UiT The Arctic University of Norway, 9037 Tromsø, Norway; 2Department of Pathology, University Hospital of North Norway, 9019 Tromsø, Norway; 3Department of Community Medicine, UiT The Arctic University of Norway, 9037 Tromsø, Norway

**Keywords:** cervical cancer screening, triage, HPV DNA test, HPV mRNA test, CIN3+

## Abstract

Background: A specific, cost-effective triage test for minor cytological abnormalities is essential for cervical cancer screening among younger women to reduce overmanagement and unnecessary healthcare utilization. We compared the triage performance of one 13-type human papillomavirus (HPV) DNA test and one 5-type HPV mRNA test. Methods: We included 4115 women aged 25–33 years with a screening result of atypical squamous cells of undetermined significance (ASC-US) or low-grade squamous intraepithelial lesions (LSIL) recorded in the Norwegian Cancer Registry during 2005–2010. According to Norwegian guidelines, these women went to triage (HPV testing and repeat cytology: 2556 were tested with the Hybrid Capture 2 HPV DNA test, which detects the HPV types 16, 18, 31, 33, 35, 39, 45, 51, 52, 56, 58, 59, and 68; and 1559 were tested with the PreTect HPV-Proofer HPV mRNA test, which detects HPV types 16, 18, 31, 33, and 45). Women were followed through December 2013. Results: HPV positivity rates at triage were 52.8% and 23.3% among DNA- and mRNA-tested women (*p* < 0.001), respectively. Referral rates for colposcopy and biopsy and repeat testing (HPV + cytology) after triage were significantly higher among DNA-tested (24.9% and 27.9%) compared to mRNA-tested women (18.3% and 5.1%), as were cervical intraepithelial neoplasia grade 3 or worse (CIN3+) detection rates (13.1% vs. 8.3%; *p* < 0.001). Ten cancer cases were diagnosed during follow-up; eight were in DNA-tested women. Conclusion: We observed significantly higher referral rates and CIN3+ detection rates in young women with ASC-US/LSIL when the HPV DNA test was used at triage. The mRNA test was as functional in cancer prevention, with considerably less healthcare utilization.

## 1. Introduction

Cytology-based screening has reduced the incidence and mortality of cervical cancer in countries with organized screening programs [1]. However, because cytology-based screening has low sensitivity in detecting high-grade lesions (cervical intraepithelial neoplasia grade 3 or worse, CIN3+) [2], several countries have replaced it with human papillomavirus (HPV) test-based screening [3,4,5]. Organized cervical screening programs aim to obtain the best possible balance between reducing the risk of cervical cancer and limiting over-management [6]. European guidelines recommend HPV test-based screening at 5-year intervals for women aged 30–60 years [7], and several countries follow these guidelines [3,8], though some have targeted screening populations of different ages [4,9]. In 2015, Norway implemented HPV DNA testing at 5-year intervals in selected counties to screen women aged 34–69 years [10] but continued to use cytology at 3-year intervals to screen women aged 25–33 years [10]. Since 2023, HPV testing is also applied for women under 34 years [11].

Indeed, HPV test-based screening for women aged 30 years or younger is not cost-effective, given the high prevalence of HPV infection [12] and low incidence of high-grade cervical lesions in this age group [13,14]. The global HPV prevalence in women aged 25–34 years was reported to be 13.9% [12], compared to 32% in women younger than 30 years in Norway [15]. Moreover, most young women clear their HPV infections within 1 (70%) or 2 years (91%) [16], so cytology-based rather than HPV test-based screening is considered better for young women [17]. Still, the proportion of women with abnormal cytology results in this age group is quite high, making good management strategies essential [18]. Compared to women with normal cytology, women with minor cytological abnormalities have a higher risk of high-grade dysplasia [19]. However, as most minor cytological abnormalities regress spontaneously [20], careful triage is crucial to avoid unnecessary referrals and healthcare utilization. The referral of all women with cytology results of atypical squamous cells of undetermined significance (ASC-US) or low-grade squamous intraepithelial lesions (LSIL) would result in overmanagement and overtreatment [21].

In a previous study, the reported 5-year risks of CIN3+ for women with screening results of ASC-US and LSIL were 2.6% and 5.2%, respectively [22]. Moreover, the progression rates from normal cytology and ASC-US/LSIL to CIN3 vary by HPV type, being faster for HPV16 than for HPV18, 31, 33, 45, and other oncogenic types [23]. This may make younger women particularly prone to the overtreatment of CIN [21], as a positive HPV test in this age group could trigger a referral and treatment process for infections that would otherwise have cleared spontaneously [20]. The degree of overtreatment will depend upon the number of HPV types targeted by the applied HPV tests, as well as other test properties [23].

It has been shown that a 5-type HPV mRNA test has a higher clinical specificity and positive predictive value than a 14-type HPV DNA test in the triage of women with minor cytological abnormalities at screening [24]. In the present study, we compared the performance of two triage tests—one 13-type HPV DNA test and one 5-type HPV mRNA test—among women aged 25–33 years with screening results of ASC-US or LSIL.

## 2. Materials and Methods

### 2.1. Data Source and Study Population

Data were obtained from the Norwegian Cervical Cancer Screening Programme (NCCSP), a division of the Cancer Registry of Norway (CRN). All cytology and pathology departments in Norway must report their results to the CRN, where results are classified according to the Bethesda System [25] and the World Health Organization dysplasia nomenclature [26], respectively. There were 47,705 women with ASCUS/LSIL registered in the NCCSP database during the time of the study. The 11-digit personal identification number assigned to all persons in Norway at birth or immigration was used to merge NCCSP data with that from other databases of the CRN.

### 2.2. Study Sample and Screening Algorithm

In the present analysis, we aimed to assess the performance of HPV DNA testing (using Hybrid Capture II) and HPV mRNA testing (using PreTect HPV-Proofer) in a specific subpopulation of women in Norway. Our study sample consisted of women aged 25–33 years who underwent cervical cancer screening between 2005–2010 and had a recorded screening result of ASC-US or LSIL in the NCCSP. Inclusion in the study sample required the absence of a previous diagnosis of CIN1+ or HSIL. Pragmatically, we identified 4115 women who met these criteria, thus forming our final study sample. Subsequently, we followed their medical records until 31 December 2013. This population was of interest as, unlike women aged 34–69 years and older who have been screened using HPV testing since 2015, women aged 25–33 years underwent cytological screening, and in cases with minor cytological abnormalities, HPV testing was used for triage. At the end of the post-screening follow-up period, we defined women with indication for colposcopy and biopsy or repeat testing (HPV and cytology) without it being performed as women with incomplete follow-up.

The screening algorithm that was in effect in Norway during the study period recommended that women with screening results of ASC-US/LSIL attend triage within 6–12 months thereafter; triage consisted of both HPV testing and repeat cytology. Depending on the triage results, the screening algorithm recommended either further follow-up or a return to the 3-year screening interval without further follow-up. To reflect best practices, we categorized the post-triage study sample into three screening algorithm-recommended (SAR) groups: SAR referral for colposcopy and biopsy, SAR referral for repeat testing (HPV and cytology), or SAR return to the 3-year screening interval (i.e., return to routine screening, Figure 1). However, actual clinical practice may deviate from the recommended practice. To increase study power, we expanded the window for triage to 3–18 months after screening results. 

### 2.3. HPV Testing

Data on the type of HPV test used for triage in different laboratories were taken from the NCCSP database. The type of HPV test was not randomized; laboratories in each geographical region made their own decisions about which commercial HPV test to use. When women with minor cytological abnormalities were triaged using conventional or liquid-based cytology (LBC), extra specimens were collected and placed in preservation and transport media for DNA (digene Specimen Transport Medium [27]) and mRNA tests (PreTect TM [28]). We confined our analyses to data reported from laboratories that used either Hybrid Capture 2 (HC2, Qiagen), an HPV DNA test that detects 13 HPV types (16, 18, 31, 33, 35, 39, 45, 51, 52, 56, 58, 59, and 68) [29], or PreTect HPV-Proofer (PreTect AS, Klokkarstua, Norway), an mRNA test that genotypes the E6/E7 full-length mRNA transcripts of five HPV types (16, 18, 31, 33, and 45) [30]. The qualitative assays were based on real-time, nucleic acid sequence-based amplification (NASBA) technology, targeting full-length E6/E7 transcripts, and included an intrinsic sample control to ensure sample adequacy [31]. Positive and negative assay controls corresponding to the viral mRNA for all targets validate the results reported by PreTect Analysis Software (https://www.pretect.no/pretecthpvprooferorg, available online: 22 February 2023). Total nucleic acids were isolated from 1 mL of the leftover LBC material preserved in ThinPrep using PreTect X (PreTect AS, Klokkarstua, Norway) and subsequently analyzed for mRNA expression according to the manufacturer’s instructions. The HPV DNA status of the specimens was detected by the hc2 HPV kit following the manufacturer’s protocols [29]. The HC2 reports only positive/negative results based on positivity for at least one included HPV type. Consequently, we report positive/negative HPV results for both DNA and mRNA tests.

### 2.4. Outcomes and Follow-Up

The primary study outcomes were HPV positivity rates at triage, SAR referral rates for colposcopy and biopsy or repeat testing after triage, and CIN3+ detection rates from triage to the end of follow-up (31 December 2013). We also explored CIN3+ detection rates at 42 (one screening interval: 36 months + 6 months) and 78 months (two screening intervals: 72 + 6 months) post-screening. We compared these outcomes between women who received HPV DNA and HPV mRNA testing at triage.

### 2.5. Statistical Analyses

All analyses were conducted in SPSS 29.0. Chi-square tests were performed for categorical variables, with a *p*-value < 0.05 as the significance level. In order to compare the age distributions among HPV DNA-tested and HPV mRNA-tested women, we divided women into age groups of 25–29 and 30–33 years old.

## 3. Results

Of the 4115 included women, 62.1% (2556/4115) received HPV DNA testing and 37.9% (1559/4115) received HPV mRNA testing at triage. The distributions for age, screening result (ASC-US or LSIL), and the most recent cytology result before screening are reported for DNA-tested and mRNA-tested women in Table 1.

The HPV positivity rate at triage among DNA-tested women was more than twice that of mRNA-tested women (52.8% vs. 23.3%; *p* < 0.001) (Table 2). Moreover, the SAR referral rate for colposcopy and biopsy after triage was significantly higher among DNA-tested women compared to mRNA-tested women (24.9% vs. 18.3%; *p* < 0.01), a pattern that was also observed for the SAR referral rate for repeat testing after triage (27.9% vs. 5.1%; *p* < 0.001). Consequently, according to the screening algorithm, more mRNA-tested than DNA-tested women should have returned to routine screening (76.7% vs. 47.2%; *p* < 0.01) (Table 2).

The CIN3+ detection rate at 42 months post-screening among women with SAR referrals for colposcopy and biopsy after triage was similar in DNA-tested and mRNA-tested women (31.7% vs. 30.5%; *p* = 0.72) (Table 3). This similarity remained when looking at the CIN3+ detection rates at 78 months post-screening (33.4% vs. 32.3%; *p* = 0.73), as there were few cases of CIN3+ diagnosed at 42 months (Table 3).

Eight cases of cervical cancer (5 squamous cell carcinomas and 3 adenocarcinomas) had been diagnosed at 78 months post-screening among DNA-tested women, while only two squamous cell carcinomas had been diagnosed among mRNA-tested women. Two of the DNA-tested women and one mRNA-tested woman most likely had false-negative HPV results at triage (data not shown).

Although there were only marginal differences in the detection rate of CIN3+ at 42 and 78 months post-screening (Table 3), different HPV positivity rates at triage led to large differences in SAR referral rates for colposcopy and biopsy and repeat testing after triage, resulting in an overall CIN3+ detection rate that was significantly higher among DNA-tested (13.1%; 345/2556) compared to mRNA-tested women (8.3%; 129/1559) at 42 months post-screening (*p* < 0.001). The corresponding values at 78 months post-screening were 14.6% (372/2556) and 9.4% (147/1559) (*p* < 0.001). The proportion of both DNA-tested and mRNA-tested women with incomplete follow-up decreased from 42 to 78 months post-screening; accordingly, the proportion of women that returned to routine screening increased among all SAR groups after triage (Table 3). Our findings showed a 6-year CIN3+ risk of 2.8% among women with triage results of ASC-US/LSIL and a negative HPV mRNA test, versus 1.4% among women with triage results of ASC-US/LSIL and a negative HPV DNA test (Table 3).

## 4. Discussion

In this study, we evaluated the background characteristics of women who underwent HPV DNA testing (using Hybrid Capture II) and those who underwent HPV mRNA testing (using PreTect HPV-Proofer). We assessed factors that could impact the outcomes of the tests, including the age distribution, the distribution of ASC-US and LSIL results in the screening process, and the most recent cytology results before screening occurred. Our analysis indicated a high level of homogeneity between the two groups, thus reducing the risk of selection bias. The HPV DNA test detects 13 HPV types, whereas the HPV mRNA test only detects 5 types, leading to a higher HPV positivity rate among the DNA-tested women. In accordance with the Norwegian screening algorithm, this higher positivity rate resulted in a higher referral rate for colposcopy/biopsy and repeat testing for the DNA-tested women, leading to a higher overall rate of CIN3+ detection. However, this came at a cost of significantly more health resources, as measured by the number of follow-up visits and colposcopy/biopsy procedures. Our findings reveal that despite the higher resource utilization and CIN3+ detection rate in the DNA-tested women, the HPV DNA test was not more effective in preventing cervical cancer, which was the primary goal of the screening.

### 4.1. HPV Positivity Rates at Triage

In our low-risk population of women aged 25–33 years with ASC-US/LSIL, the HPV positivity rate at triage among DNA-tested women was lower (52.8%) than the global rate observed among women of all ages with ASC-US/LSIL who received this triage test (59.4%) [32]. The HPV positivity rate observed when a 5-type HPV mRNA test was used in the triage of Danish women aged 23–39 years with LSIL (34.7%) [33] was higher than the rate we recorded among our mRNA-tested women (23.3%). This could be because only women with LSIL were included in the Danish study, as HPV prevalence is generally higher in women with LSIL than those with ASC-US [32]. Another Danish study of women under 30 years of age with ASC-US/LSIL reported HPV positivity rates at triage for DNA- (any assay), 14-type mRNA-, and 5-type mRNA-tested women of 82.5%, 73.5%, and 40%, respectively [21]. However, HPV prevalence tends to decrease with age [12]; therefore, the higher positivity rate in the Danish study compared to ours might be due to the inclusion of women who were younger than those in our study population. 

### 4.2. Referral Rates for Colposcopy and Biopsy and Repeat Testing after Triage

According to Norwegian guidelines, significantly more DNA-tested women in our study sample were to be referred for colposcopy and biopsy and repeat testing. In the Danish study, which included women under 30 years of age with ASC-US/LSIL, biopsy rates in DNA- (67%), 14-type mRNA- (77%), and 5-type mRNA-tested women (58%) [21] were generally higher than the SAR referral rates for colposcopy and biopsy we observed in our study. However, the Danish referral rate lessened as the HPV types included in the assay decreased.

### 4.3. CIN3+ Detection Rate

As a consequence of the higher SAR referral rates for colposcopy and biopsy and repeat testing, DNA-tested women had a significantly higher overall CIN3+ rate compared to mRNA-tested women at 42 months post-screening. Our results support the general conclusions that, in the triage of women with minor cytological abnormalities, an HPV mRNA test has lower referral rates than an HPV DNA test [24,34,35]. However, compared to the 14-type HPV DNA test, the reliability of the 5-type HPV mRNA test to rule out CIN3+ among women who test negative has been considered too low to be used as a basis for the determination to return to routine screening. Considering the 6-year CIN3+ risk of 2.8% among women with triage results of ASC-US/LSIL and a negative HPV mRNA test versus 1.4% among women with triage results of ASC-US/LSIL and a negative HPV DNA test in our study—and based on the principle of “equal management for equal risk”, which guides patient management in screening—our results suggest that women who are HPV-negative at triage require a further 1-year surveillance period, in order to respect the accepted 5-year CIN3+ risk threshold (>0.55 <5.0%), as the 5-year CIN3+ risk rate is 2.6% for ASC-US alone [22,36,37]. In accordance with a model-based economic evaluation analysis of the triage of young adult women with minor cervical lesions [38], SAR referral for colposcopy and biopsy was twice as high among our DNA-tested women as our mRNA-tested women. Thus, compared with the mRNA test, the usage of the DNA test at triage increased the workload of gynecologists and laboratories by more than double.

### 4.4. HPV Types Included in the Test

The number of HPV types included in the test may influence positivity, referral, and CIN3+ detection rates. A previous report showed that only 5–6 HPV types (16, 18, 31, 33, 45, or 52) were present in 85% of invasive cervical cancer cases, while the other eight HPV types included in 13–14 type HPV DNA tests were detected just in 1.5% of invasive cervical cancer cases [39]. Thus, the 5-fold higher SAR referral rate for repeat testing that we observed among DNA-tested women compared to mRNA-tested women may be attributed to the higher number of oncogenic HPV types included in the DNA test. This observation was also made in a Danish study, in which a 14-type HPV mRNA test yielded higher positivity rates than a 5-type HPV mRNA test [33]. The influence of the number of HPV types included in the mRNA test on both positivity rates and referral rates was also observed in the other Danish study of women under 30 years of age with ASC-US/LSIL [21]; when the 14-type and 5-type HPV mRNA tests were compared, the positivity rate decreased from 73.5% to 40% and the biopsy rate decreased from 77% to 58% [21]. Moreover, generally, when more types are included in the HPV test, the sensitivity for CIN2+/CIN3+ increases, but the specificity for CIN2+/CIN3+ decreases. A 14-type HPV DNA test (Cobas) showed higher sensitivity to detect CIN2+ than a 5-type HPV mRNA test (Proofer) in the triage of women with ASC-US/LSIL (100% vs. 79%), while its specificity was lower (84% vs. 91%) [34]. This reduction in specificity with an increasing number of HPV types is more visible with the decreasing age of the study population. A 14-type HPV mRNA test (Aptima) showed higher sensitivity to detect CIN2+ in the triage of women with LSIL aged 23–65 years compared to a 5-type HPV mRNA test (Proofer) (94% vs. 77%), while its specificity was considerably lower (34% vs. 69%) [33]. Although the gap in test sensitivity (93% vs. 80%) did not change substantially, the gap in specificity (19% vs. 64%) widened among women aged 23–29 years [33].

### 4.5. Overmanagement and Overtreatment

The Norwegian health care system has allocated limited resources for colposcopy and biopsy examinations [40]; therefore, the risk of overmanagement and overtreatment is of great concern in the NCCSP. The number of cervical cancers has increased from 311 in 2012 to 340 in 2021 in Norway, and the incidence rate showed an increasing trend (from 12.4/10^5^ in 2010 to 14.5/10^5^ in 2019) before decreasing slightly to 12.6/10^5^ in 2021 [18].

The higher overall CIN3+ detection rate among our DNA-tested women compared to mRNA-tested women was attributable to significantly higher healthcare utilization in the former group, as measured by the number of SAR referrals for colposcopy and biopsy or repeat testing. Overall, 345 DNA-tested women (13.1%) had been diagnosed with CIN3+ at 42 months post-screening, with 17 more diagnosed at 78 months post-screening. Among mRNA-tested women, 129 (8.3%) had been diagnosed with CIN3+ at 42 months, and 33 more at 78 months, post-screening. This revealed the burden of overmanagement among DNA-tested women, as 216 (345 − 129 = 216) more DNA-tested women with CIN3+ received treatment to avoid 16 (33 − 17 = 16) more women having CIN3+ results within 36 months, i.e., before the next screening round.

The number of treatments using the loop electrical excision procedure (LEEP) in Norway has increased from 3743 in 2010 to 7354 in 2021 without any reduction in the number of women with cervical cancer [18]. A low SAR referral rate for colposcopy and biopsy will reduce healthcare costs, as well as the use of LEEP treatment and its negative impacts (e.g., unfavorable pregnancy outcomes, such as preterm birth, spontaneous abortion, and psychological stress [41]). 

Not all CIN3 cases, and only a minority of CIN2 cases, will progress to cancer [42,43,44]. Research has shown that only 30% of large CIN3+ lesions will progress to cervical cancer during 30 years if left untreated [43], meaning that a substantial proportion of women with these conditions will undergo biopsy, and potentially LEEP treatment, unnecessarily. The purpose of the NSSCP is not to find as many CIN3+ as possible but to prevent as many cancer cases as possible and simultaneously balance benefits and harms. Even though more DNA-tested women in our study received LEEP treatment, eight cases of cervical cancer appeared among them during the study period, while only two cancer cases were diagnosed among mRNA-tested women. This illustrated the possibility of preventing more cancer cases and avoiding overtreatment by using a more specific triage test and targeting women with the highest cancer risk [34].

### 4.6. Strengths

The importance of this study lies in the long-term comparison of the predictive and performative value of HPV DNA and HPV mRNA tests in the triage of a large number of 25–33-year-old women with minor cytological abnormalities at screening. We used the NCCSP database, a nationwide, register-based platform embedded within the CRN, which allowed us to identify women with minor cytological abnormalities in Norway. Moreover, we used the 11-digit personal identification number assigned to all persons at the time of birth or immigration in Norway to merge data on cytology results, histological results, cancer cases, etc.

Although ours is not a randomized study, we consider the similarity in the characteristics of our DNA- and mRNA-tested women (Table 1) as an advantage, because it mimics the condition of randomized selection. Another unique advantage of our study is that we followed all women for up to 6 years after triage.

### 4.7. Limitations

Our analysis assumed that women, general practitioners, and gynecologists consistently followed the screening algorithm perfectly; however, in real clinical practice, there are probably more referrals for cytology/biopsy and repeat testing. Therefore, we also investigated the CIN3+ detection rate among women who were returned to screening (Table 3), to reflect the reality that some women are referred for colposcopy and biopsy even when such a referral is not recommended in the screening guidelines. The proportions of women with incomplete follow-up decreased during the study period. However, the proportions of women who did not attend and women with incomplete follow-up remained considerably high, especially among women with SAR referral for routine screening (63–71.9% at 78 months post-screening). This research was conducted in Norway, a country with the highest human development index between 2005–2010 and free public access to essential healthcare services. With a generally high socio-economic status and high coverage for vaccination and screening, it should also be expected that women would have a high rate of adherence to follow-up.

## 5. Conclusions

In this study, we evaluated the performance of HPV DNA testing (using Hybrid Capture II) and HPV mRNA testing (using PreTect HPV-Proofer) in a population of young women with ASC-US/LSIL. We found that when the HPV DNA test was used for triage, there was a significantly higher rate of referral for colposcopy and biopsy, repeat testing, and CIN3+ detection. However, this did not result in improved cancer prevention compared to the mRNA test. The mRNA test demonstrated similar efficacy in cancer prevention while requiring significantly less healthcare utilization. Based on these findings, healthcare authorities considering a triage test for minor cytological abnormalities among younger women may prefer the mRNA test to minimize overmanagement and reduce unnecessary healthcare utilization.

## Figures and Tables

**Figure 1 ijerph-20-04119-f001:**
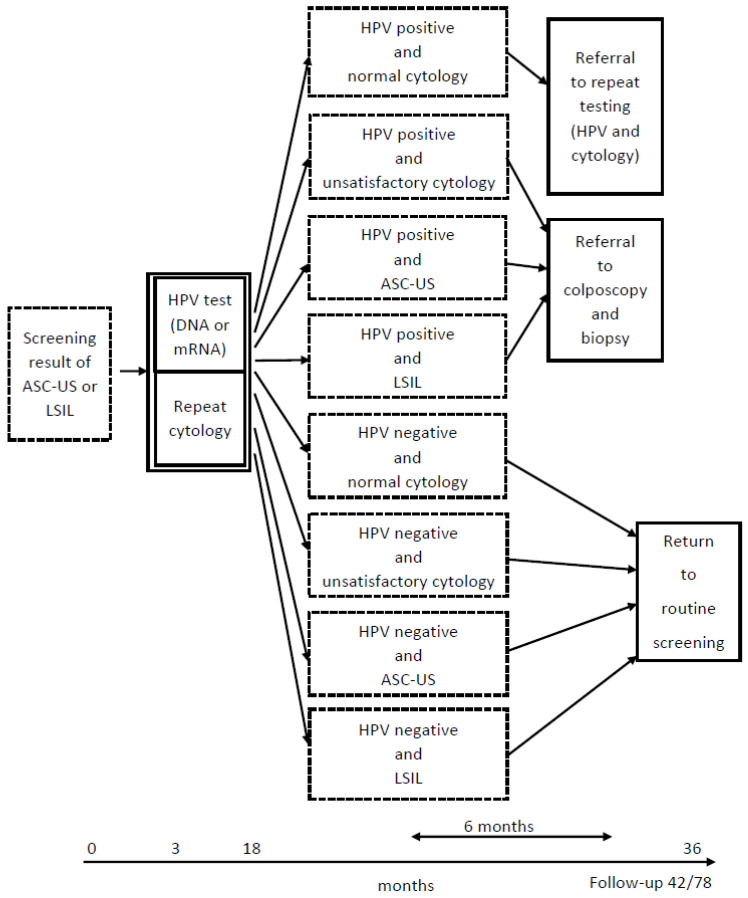
Screening algorithm in Norway during the study period. HPV, human papillomavirus; ASC-US, atypical squamous cells of undetermined significance; LSIL, low-grade squamous intraepithelial lesions; return to routine screening, return to the 3-year screening interval without further follow-up.

**Table 1 ijerph-20-04119-t001:** Comparison of characteristics among HPV DNA-tested and HPV mRNA-tested women.

		HPV DNA *n* = 2556 (%)	HPV mRNA *n* = 1559 (%)	*p*-Value
**Age (years)**	25–29	60.4	59.9	0.771
	30–33	39.6	40.1	
**Screening result**	ASC-US	65.9	68.8	0.06
	LSIL	34.1	31.2	
**Most recent cytology result before screening**	No Previous Test	35.5	36.0	0.603
	Unsatisfactory	3.0	3.5	
	Normal	61.5	60.5	

Abbreviations: HPV, human papillomavirus; ASC-US, atypical squamous cells of undetermined significance; LSIL, low-grade squamous intraepithelial lesion.

**Table 2 ijerph-20-04119-t002:** HPV and cytology results at triage among HPV DNA-tested and HPV mRNA-tested women.

Triage Results		HPV DNA *n* = 2556 (%)	HPV mRNA *n* = 1559 (%)
HPV	Cytology		
Positive	Normal	713 (27.9%)	79 (5.1%)
	Unsatisfactory	32 (1.3%)	3 (0.2%)
	ASC-US	285 (11.2%)	131 (8.4%)
	LSIL	320 (12.5%)	151 (9.7%)
Total		1350 (52.8%)*	364 (23.3%) *
Negative	Normal	1042 (40.8%)	852 (54.7%)
	Unsatisfactory	43 (1.7%)	26 (1.7%)
	ASC-US	93 (3.6%)	206 (13.2%)
	LSIL	28 (1.1%)	111 (7.1%)
Total		1206 (47.2%) *	1195 (76.7%) *

* *p* < 0.001. Abbreviations: HPV, human papillomavirus; ASC-US, atypical squamous cells of undetermined significance; LSIL, low-grade squamous intraepithelial lesion.

**Table 3 ijerph-20-04119-t003:** Status and CIN3+ detection rates at 42 and 78 months post-screening by the screening algorithm-recommended (SAR) group after triage among HPV DNA-tested and HPV mRNA-tested women.

SAR Group after Triage	Status	42 Months Post-Screening		*p*-Value	78 Months Post-Screening		*p*-Value
		HPV DNA	HPV mRNA		HPV DNA	HPV mRNA	
SAR Referral for Colposcopy and Biopsy		* n * = 637 (%)	* n * = 285 (%)		* n * = 637 (%)	* n * = 285 (%)	
	Did not attend	7 (1.1)	1 (0.4)		7 (1.1)	1 (0.4)	
	Incomplete follow-up	292 (45.8)	113 (39.6)		135 (21.2)	45 (15.8)	
	Returned to routine screening	86 (13.5)	38 (13.3)		229 (35.9)	99 (34.7)	
	CIN2	50 (7.8)	46 (16.1)		53 (8.3)	48 (16.8)	
	CIN3+	202 (31.7)	87 (30.5)	0.72	213 (33.4)	92 (32.3)	0.73
SAR Referral for Repeat Testing (HPV and Cytology)		*n* = 713 (%)	*n* = 79 (%)		*n* = 713 (%)	*n* = 79 (%)	
	Did not attend	37 (5.2)	3 (3.8)		37 (5.2)	3 (3.8)	
	Incomplete follow-up	376 (52.7)	45 (57.0)		181 (25.4)	17 (21.5)	
	Returned to routine screening	139 (19.5)	8 (10.1)		322 (45.2)	34 (43.0)	
	CIN2	29 (4.1)	3 (3.8)		31 (4.3)	3 (3.8)	
	CIN3+	132 (18.5)	20 (25.3)	0.145	142 (19.9)	22 (27.8)	0.099
SAR Return to Screening		*n* = 1206 (%)	*n* = 1195 (%)		*n* = 1206 (%)	*n* = 1195 (%)	
	Did not attend	308 (25.5)	241 (20.2)		308 (25.5)	241 (20.2)	
	Incomplete follow-up	799 (66.3)	819 (68.5)		560 (46.4)	511 (42.8)	
	Returned to routine screening	87 (7.2)	103 (8.6)		319 (26.5)	398 (33.3)	
	CIN2	1 (0.1)	10 (0.8)		2 (0.2)	12 (1.0)	
	CIN3+	11 (0.9)	22 (1.8)	0.05	17 (1.4)	33 (2.8)	0.02

Abbreviations: HPV, human papillomavirus; CIN2, cervical intraepithelial neoplasia grade 2; CIN3+, cervical intraepithelial neoplasia grade 3 or worse; return routine to screening, return to the 3-year screening interval without further follow-up.

## Data Availability

Not applicable.

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
