# Peer review of "13-Type HPV DNA Test versus 5-Type HPV mRNA Test in Triage of Women Aged 25–33 Years with Minor Cytological Abnormalities–6 Years of Follow-Up"

_ijerph, 2023, doi:10.3390/ijerph20054119_

Round 1
Reviewer 1 Report
This study with a large sample size is well detailed and understandable. I have made some comments in the PDF because some methodological clarifications are necessary.

Author Response
Dear reviewer,
Thank you very much for taking time to review our manuscript.
We really appreciate your comments in improving our paper. We provided our responses to your comments in a separate Word file attached to this letter.
Thank you again for your time and consideration and your valuable suggestions in improving our paper. We hope our updates were fulfilled your concerns.
Best regards,
Rad et. al.

Reviewer 2 Report
Overall, the whole structure of this study is good and some corrections are recommended for providing clear information. Particularly, I listed the following comments in detail here.
Major concerns:
In the abstract, the author needs to mention the ingredients of methods, and materials. Names and terms should be completely mentioned for the first time. Also, the finding of the assay could be added step by step based on material and method. I recommend considering regular assays and results.
In the introduction, some sentences lack references. For example, “However, because cytology-based screening has low sensitivity to detect high-grade lesions (cervical intraepithelial neoplasia grade 3 or worse, CIN3+) [2], several countries have replaced it with human papillomavirus (HPV) test-based screening. O”, “This may make younger women particularly prone to overtreatment of CIN, as a positive HPV test in this age group could trigger a referral and treatment process for infections that would otherwise have cleared spontaneously. The degree of overtreatment will depend upon the number of HPV types targeted by the applied HPV tests, as well as other test properties”, and so on.
In methods, the author needs to mention the ingredients of methods, locations, and materials. Please add references for all tests.
In the discussion, discuss your results before relating them to the results of other published work. Also, the author must step by step to come to the results and comparison with others.
What is your conclusion? Do the authors have more thoughts on this field?
Author Response

(The authors gave the same response as above.)

Reviewer 3 Report
The authors using surveillance data to compare the performance of HPV DNA test to mRNA test.
· Line167-170 (Table 3), what was the definition of incomplete follow-up? Why were the incomplete follow-up rates higher at 42 months than at 78 months? Why did the precents for people who did not attend for DNA and mRNA were the same at 42 months and 78 months for all triages? Please include both number and percent in the tables (have the percentage in paratheses as label) for clarity.
· Line 177-190, were the positivity rates using DNA and mRNA from this study lower than other studies because of the underlying infection prevalence and different performance of tests used (i.e., sensitivity and specificity)?
· Line 210-214, please further clarify. Do the authors suggest to change the current surveillance monitoring to 6 years? Were the study results compared to the 5 yrs results (published or historical data)?
· Line 216-218, although referrals to colposcopy and biopsy were higher in DNA-tested women, the overall CIN3+ detection rate was also significantly higher in DNA-tested than mRNA-tested women as the authors stated (line 164-166).
· Line 232-235, were the sensitivity and specificity here referred to the outcome of cervical cancer, CIN2+ detected or HPV infections?
· Line 232-234, the statement is questionable. Based on the comparison presented here, the sensitivity difference was 21% and the specificity difference is 7%. How about hybrid capture performance? Were the sensitivity and specificity of Cobas and Hybrid Capture similar? The positive and negative predicted values also depend on the underlying prevalence.
· Line 252-258, CIN3+ patient can be treated with cryotherapy, laser therapy, etc. to prevent cancer development. What was the reason to consider the excess number CIN3+ detected using DNA-test as burden of over-management in Norway?
· Line 260-264, when the number of women went through excision procedure did not increase, would the number of cervical cancer cases increase? As I know, cervical cancer vaccine trials do not use cervical cancer as the outcome because cervical cancer can take years to be observed.
· Line 266-274, although I agree with author’s point to identify more cervical cancer not CIN3+, however, CIN3+ is highly associated with the development of cervical cancer. Based on the study results, 0.3% (8/2556) vs 0.1% (2/1559) cervical cancer were found among DNA and mRNA group, respectively. Also, overall CIN3+ detection rate in DNA-tested group were significantly higher than mRNA-tested group. How about the price difference between these two tests in Norway?
· Line 298-299, the conclusion seems questionable for the questions mentioned above.
Author Response

(The authors gave the same response as above.)

Reviewer 4 Report
Rad et al present how different methods of detecting HPV genetic material leads to identification of women who develop HPV-related disease. This is in comparison to the cytology that has been traditionally used to detect abnormal cell from the female reproductive tract. The overall representation of different outcomes and age groups with the total number of patients is excellent. The paper is well written and thought out.
Comments:
1) Are you able to determine the outcome of those individuals that developed CIN3+ and does the screening help improve patient outcomes?
2) Do you happen to have a quantitative number of HPV DNA or RNA and determine if there is a correlation between genetic material and disease outcome?
3) In your discussion, you bring up the impact of overmanagement and overtreatment of patients. But screening on this scale is not universal across the developed world and while strengths and limitations are discussed, what are patient outcomes/quality of life. Could you at least discuss the value of these kind of evaluations in regards to patients and why it is of value to them?
Author Response

(The authors gave the same response as above.)

Round 2
Reviewer 1 Report
Dear editors,
thanks to the authors for their additions and corrections. The article is in my opinion publishable in this form
Author Response
Dear reviewer,
Thank you very much for your time and consideration to review our paper and providing constructive comments to improve it.
We are pleased to receive your positive opinion.
Best regards,
Rad et al
Reviewer 3 Report
The authors using surveillance data to compare the performance of HPV DNA to mRNA testing strategies. The main concern is that the study results do not seem to support the study conclusion. The authors might want to consider better distinguish evaluating testing performance (detection ability) and cost-effectiveness; and be more careful about interpretations of the statistical results.
· As author mentioned, the primary goal of the screening is to detect CIN3+ for cervical cancer prevention. In the result section (Line 173-177, Table 3), the CIN3 detection rates were 13.4% (345/2256) and 8.2% (129/1559) at 42 months, for HPV DNA and mRNA testing strategy, respectively; and 14.6% (372/1559) and 9.4 (147/1529) at 78 months. The CIN3+ detection rate for DNA test were 5% higher than mRNA test at both time points. The differences were also statistically significant (line 187-195).
· In addition, the results (line 178-181), there were 8 (0.3%) and 2 (0.1%) cervical cancers found for using DNA testing and mRNA testing, respectively. This result suggested that the DNA testing detected more cervical cancer during the study time period.
· Furthermore, the results indicated that there were 17 (1.4%) CIN3+ from DNA testing group and 33 (2.8%) from mRNA group found at 78 months among those who were triaged to return to routine screening (p_value=0.02). This suggested that more CIN3+ cases were missed in mRNA group initially.
· Yet, the authors concluded that HPV DNA testing were not more effective.
o Line, 216-217, the authors stated that, “HPV DNA test was not more effective in preventing cervical cancer, which was the primary goal of the screening.” (line 216-217).
o Line 346-347, the authors concluded that “DNA test did not result in improve cancer prevention compared to the mRNA test. The mRNA test demonstrated similar efficacy in cancer prevention”.
As the objective of this study is to compare the testing performance (DNA vs mRNA), these conclusions appear to contradict with the study results. Also, there were many women did not attend or incomplete follow-up (more than 70%), this might also affect the results.
Other suggestions:
Line 168, it should be 25% based on Table 3.
Line 191, 129/1559 should be 8.2%.
Line 248-250, it should be included in the Results section.
Line 254, clarify (>0.55 <5.0%). Were these economic studies conducted in Norway?
Line 294-300, this argument is questionable. Overall, there were 5% more CIN3+ detected using DNA than mRNA at 42 and 78 months.
Line, 310-311, I agreed that screening is to prevent as many cancer cases as possible. CIN3+ is associated with cancer development.
Author Response
Dear reviewer,
We would like to thank you for providing constructive and detailed review comments on our manuscript.
Your suggestions and advice have helped us to improve the quality of our manuscript.
We provided our response and the revised version of our manuscript in line with your comments and suggestions, in the attachment.
Best regards,
Rad et al
